**Brief Communication: An Ice-Debris Avalanche in the Nupchu Valley, Kanchenjunga Conservation**
**Area, Eastern Nepal**
**Alton C. Byers,[1] Marcelo Somos-Valenzuela,[2] Dan H. Shugar,[3] Daniel McGrath,[4] Mohan B. Chand,[5]**
**Ram Avtar[6]**
[1]Institute of Arctic and Alpine Research, University of Colorado at Boulder, Boulder, Colorado USA
8  80309
[2]Department of Forest Sciences, Faculty of Agriculture and Environmental Sciences, University of La
Frontera, Temuco, Chile, 4780000
[3]Water, Sediment, Hazards, and Earth-surface Dynamics (waterSHED) Lab, Department of Geoscience,
University of Calgary, 2500 University Drive NW, Calgary, Alberta, T2N 1N4
Canada
[4]Department of Geosciences, Colorado State University, Fort Collins, CO, USA, 80523
[5]Department of Environmental Sciences, Patan Multiple Campus, Tribhuvan University, Lalitpur, Nepal
[6]Faculty of Environmental Earth Science, Hokkaido University, Sapporo Japan 060-0810
**Correspondence**: Alton C. Byers (alton.byers@colorado.edu)

**Abstract** Beginning in December 2020, a series of small-to-medium, torrent-like pulses commenced
upon a historic debris cone located within the Nupchu valley, Kanchenjunga Conservation Area (KCA),
Nepal. Sometime between 16 and 21 August 2022 a comparatively large ice-debris avalanche event
occurred, covering an area of 0.6 km$^2$ with a total estimated volume of order $10^6$ m$^3$.  The area of the
debris cone left by the August 2022 event increased the historic debris cone area by 0.2 km$^2$ (total
area: 0.6 km$^2$). Although no human or livestock deaths occurred, the increase in torrent-like pulses of
debris upon this historic debris cone since 2020 exemplifies a style of mass movement that may
become increasingly common as air temperatures rise in the region.  Although the magnitude of this
event was small compared to events like the 2021 Chamoli avalanche, the widespread distribution and
frequency of comparable events presents a substantial, and potentially increasing, hazard across High
Mountain Asia.
**1 Introduction**
Large magnitude but low frequency events in the high mountains can include a variety of familiar and
poorly understood cryospheric processes, including glacial lake outburst floods (GLOFs) (Lamsal et al.
2014), snow/ice/rock avalanches (Shugar et al. 2021), landslide-induced avalanches and floods (Byers
et al. 2019), englacial conduit floods (Rounce et al. 2017), and others (see: Byers et al. 2022). Today,
enhanced communications and remote sensing technologies enable rapid identification and location
of such events, often within hours of their occurrence. Many, however, remain unreported because of
their remoteness, inaccessibility, poor communications, and/or absence of people (see: Byers et al.
2020). In this *Brief Communication,* we report on a large ice-debris avalanche that occurred sometime
between 16 and 21 August 2022 in the Kanchenjunga Conservation Area (KCA), eastern Nepal. The
event is noteworthy not only because of its probable linkages to climate change impacts in the region,
but also because local residents were unaware of its occurrence, as were the Government of Nepal
and climate change research entities in Kathmandu. Here we briefly document the event and describe
its present and future implications for local communities, scientists, and governments.

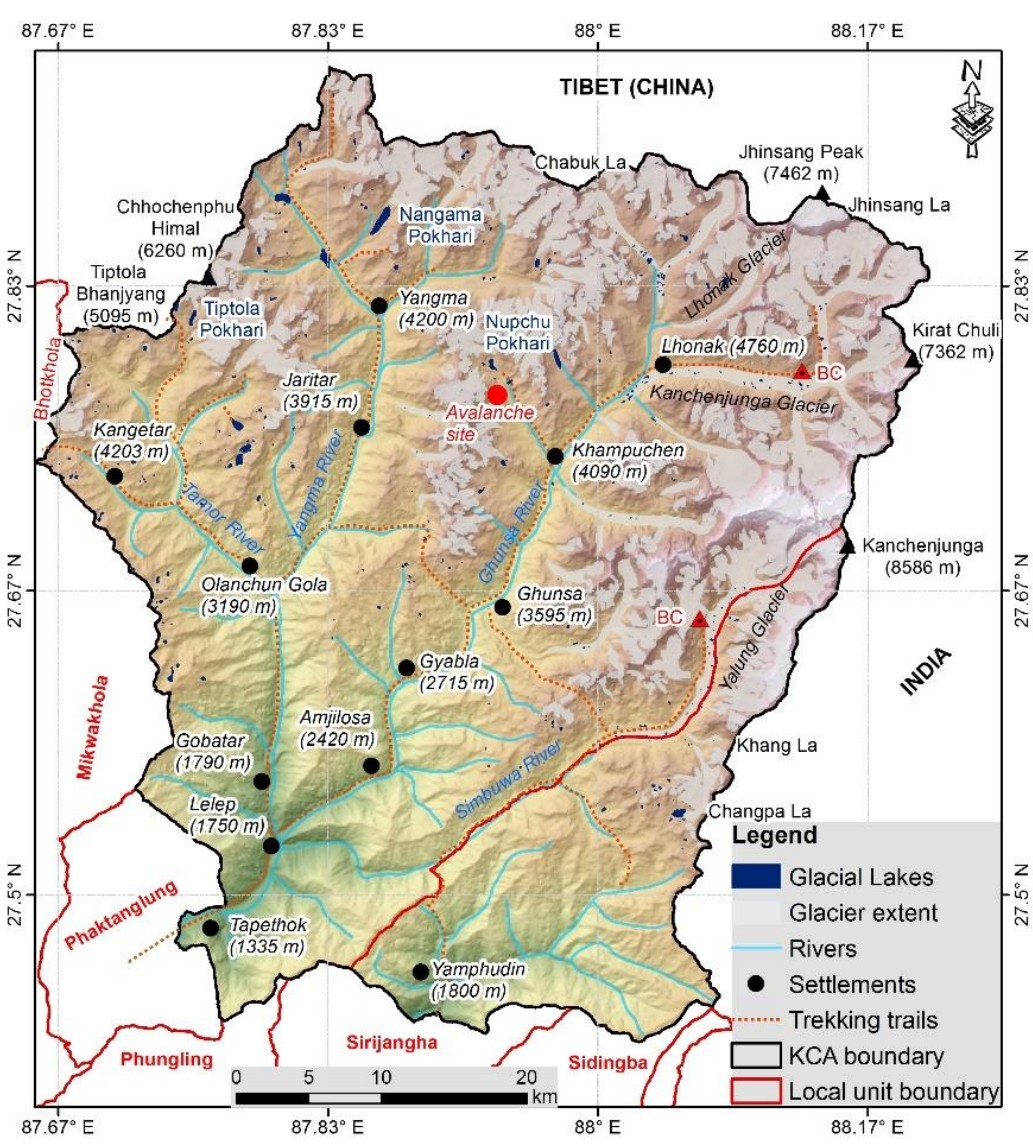

**Figure 1. Kanchenjunga Conservation Area and location of the Nupchu ice-debris avalanche.**

## 2 Setting

The KCA is a 2,035 km$^2$ protected area established in 1997 by the Nepalese Department of National Parks and Wildlife Conservation, with management responsibility handed over to local communities in 2006 (WWF Nepal 2018) (Figure 1). It is home to a range of ethnic groups of primarily Tibeto-Burman origin that include Limbu, Rai, Tamang, Gurung, Magar, Chhetri, and Sherpa (Thapa 2009). Livelihoods were traditionally based upon agriculture, livestock raising, and trade with Tibet, but globalization, outmigration, and new road construction over the past 15 years has rapidly changed the character of both the social and environmental landscape. The South Asian monsoon dominates weather patterns, with most rainfall falling between June and September (Kandel et al. 2019).

Based upon an analysis of 1962−2000 satellite imagery, valley and mountain glaciers in the KCA cover approximately 488 $\pm$ 29 km$^2$ and exhibit an overall negative glacier surface area loss of 0.5 $\pm$ 0.2% yr$^{-1}$

(Racoviteanu et al. 2015). Valley glaciers are largely debris-covered and have been receding since the
most recent maximum during the Little Ice Age.  Hooker (1854), for example, wrote in 1849 of
observing glacial moraines that provided proof "…of glaciers having once descended to from 8,000 to
10,000 feet in every Sikkim and east Nepal valley…" (Hooker 1854: 166). The British alpinist Freshfield
(1903: 236) writes of the "glacial shrinkage" he encountered in the Lhonak region in 1899, as well as
throughout both the Nepal and Sikkim sides of the Kanchenjunga massif.  Although the Kanchenjunga
region received some of the earliest study, exploration, and mountaineering expeditions in Nepal by
outsiders (Thapa 2009), relatively little glacier and cryospheric hazards research has been conducted
to date. For example, until 2019 only one GLOF event was on record for the region (Watanabe et al.
1998; ICIMOD 2011), although subsequent research revealed that at least seven others had occurred
since 1921 (Byers et al. 2020). The Nupchu valley, where the ice-debris avalanche of concern occurred,
is used seasonally for yak herding, potato farming, and tourism, with four operational tourist lodges in
the village of Kampuchen as of the fall of 2022 (Figure 1).
**3 Methods**
Field-based observations and assessments of Nupchu Pokhari (glacial lake), other nearby lakes, and
the ice-debris avalanche were conducted between 1−20 September 2022.  Methods included GPS-
based route mapping, photography of avalanche features, oral testimony, and literature reviews.
Historic (declassified KH-9 Hexagon satellite imagery; see: Maurer et al. 2019; Dehecq et al. 2020) and
recent (Planet Dove and SuperDove) satellite imagery revealed the sequence of avalanche/debris flow
events between 1975 and 2023 (Figure 2). Numerical simulations of the avalanche were conducted
using R.Avaflow version 3 (Mergili and Pudasaini 2014−2023; Mergili et al. 2017), a state of the art
software that has been used globally to study ice/rock avalanches events (Zhang 2022). Numerical
simulations were used to provide upper limit volume estimation of the avalanche, which were
constrained by field observations. We used the parameters from Zhang et al. (2022) to produce a
single-phase model scenario for three different volumes: 1, 2.5, and 5 million $m^3$. For calibration, we
modified the internal friction angle of the mixture to match the extension of the debris left by event.
For the terrain elevation, two DEMs were used: the ALOS PALSAR Radiometric Terrain Corrected high
resolution 12.5 m DEM (AP_13152_FBD_F0540_RT1) (ASF DAAC 2014) and the High Mountain Asia 8m
resolution DEM (Shean, 2017), that was void filled using the Elevation Void Fill function in ArcGIS 10.8.
**4 The Event**
The investigation of the Nupchu valley was initiated by local concerns about Nupchu Pokhari
(27.790708° N, 87.934275° E) as being one of the most dangerous glacial lakes in terms of a potential
GLOF (Figure 1). Periodic, smaller floods from the upper Nupchu valley were reported, and assumed
locally to have originated in the Nupchu Pokhari, although no supporting evidence was available. Our
field reconnaissance results of Nupchu Pokhari on 12 September 2022, however, suggested that the
lake posed a moderate risk of flooding, largely based on the absence of overhanging ice and other
potential flood triggers. This assessment corroborates the findings of Rounce et al. (2017) which
concluded that the 0.129 $km^2$ Nupchu Pokhari presented only a moderate risk of flooding because of
(a) no apparent growth between 2000 and 2015, (b) absence of avalanche pathways into the lake (i.e.,
in line with the direction of the lake and its outflow), and (c) absence of landslide pathways entering
the lake.
The August 2022 ice/debris avalanche event was unexpected. Field staff had conducted a
reconnaissance of the valley below Nupchu Pokhari in early August 2022 to check out potential
camping sites, at which time the upper valley was primarily pastureland. When the field team and A.C.
Byers returned in early September, the original path was blocked by massive ice-debris avalanche
material (27.774328° N, 87.941064° E) that had clearly occurred at some point in the interim (Figure
2). Our team and *dzopkio* (yak-cattle crossbreeds used as pack animals) were nevertheless able to
climb up and over the avalanche debris to the upper Nupchu valley, but at the time the source and
triggers related to the event remained unknown.
The original historic debris cone was found to have covered an area 0.402 km$^2$ that had been relatively
stable for at least 45 years, based upon the oldest satellite imagery available (i.e., 1975) (Dehecq et al.
2020) (Panel A, Figure 2). Time series satellite images revealed the periodic occurrence of surficial
debris flows upon this original deposition. That is, beginning in 2020, a series of small-to-medium,
torrent-like pulses commenced (Panel C through G, Figure 2), culminating in the relatively large event
that occurred sometime between 16 and 21 August 2022 (Panel H, Figure 2). The area of the debris
cone left by the August 2022 event increased the original area covered by 0.2 km$^2$ (total area: 0.6
km$^2$). Of the three different volume estimates tested (1, 2.5, and 5 million m$^3$) using two DEMs and
R.Avaflow, an avalanche volume of 1 x 10$^6$ m$^3$ using an ALOS PALSAR  RTC DEM most consistently
matched the extent (red line) and depth of the new debris cone deposited as determined by our field
observations (Figure 3).
Our team was unable to locate the event on any seismographs, most likely related to the absence of
instrumentation in this part of the eastern Himalayas. Based upon direct field observations as well as
satellite imagery, the avalanche had clearly blocked and temporarily dammed the water from the
Nupchu Khola (river) at its onset, which was nevertheless able to cut down through the ice and
sediment deposited to form a steep canyon estimated at >10 m depth. The presence of shrubs (e.g., *J.*
*indica*) fully stripped of their bark was testimony to the high velocities of the flood- and meltwater
produced by frictional forces during the event, a phenomenon reported for other rockfall-induced
landslides in Nepal (Byers et al. 2019).
**5  Discussion**
Interestingly, individuals in the community of Kampuchen, only 5 km downstream of the event, were
unaware of the avalanche. Yak herds had already returned from the high pastures to the village by
early August, the community was busy harvesting potatoes and preparing for the fall tourist season,
and no obvious changes in the Nupchu Khola had been observed (e.g., Kargel 2014 for a description of
changes in the Seti Kosi prior to the catastrophic flooding of 5 May 2012). Thus, authorities in
Taplejung and Kathmandu were also unaware of the event as of September 2022, which is typical of
many large-scale cryospheric events in remote regions of the Himalayas (e.g., Byers et al. 2022).
Still, the acceleration of torrent-like pulses of debris upon the historic debris cone since 2020 suggests
that these events may be linked to contemporary warming trends, similar to those that may have
triggered larger-scale mass wasting events elsewhere in the Himalaya (e.g., Shugar et al. 2021; Kääb et
al. 2021; Taylor et al. 2023). The frequency of such ice-debris flow events within the KCA region, and
more broadly across the Himalaya, is unknown. However, with projections of continued warming in
these regions (e.g., Lalande et al. 2021), a more systematic approach to determining their historic
frequency, as well as a better understanding of their triggers, is warranted. After further evaluation,
vulnerable villages, such as Kampuchen, may wish to consider the installation of preventative

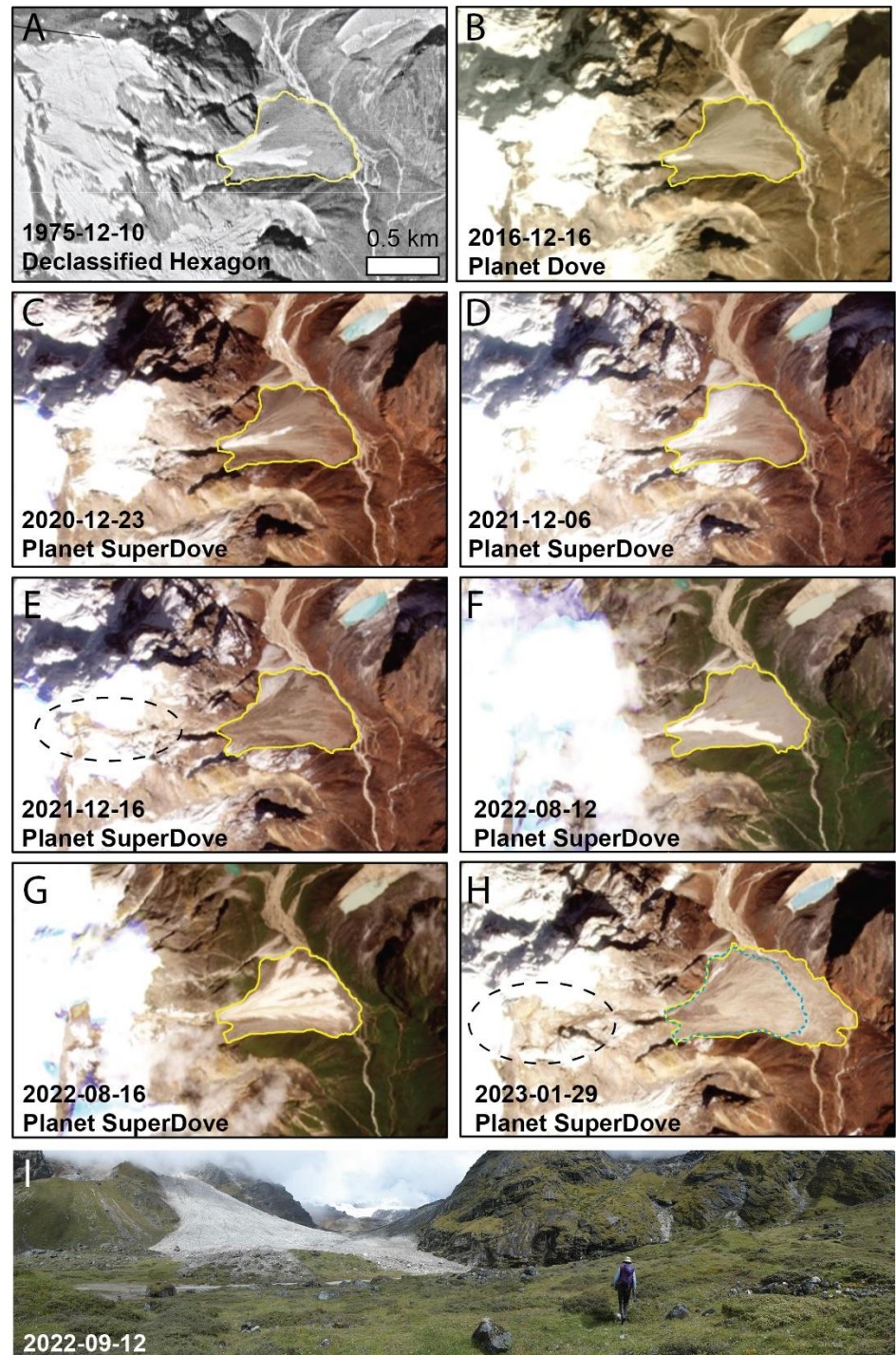

Figure 2. Time series satellite images showing the periodic occurrence of surficial debris flows upon the original deposition. These appear to have accelerated in both frequency in magnitude beginning several years ago, leading up to the main event that occurred between 16 and 21 August 2022. Blue dashed outline in panel H is the 1975 outline of the debris cone, while the dashed black circle identifies the failure zone. The photograph at the bottom provides an oblique view of the ice/debris avalanche about three weeks after it occurred (photograph by A. Byers). Panel H shows an image from early 2023, as imagery from immediately after the ice-avalanche in mid-August 2022 were partially obscured by clouds (KH-9 imagery courtesy of USGS; Planet Dove and SuperDove imagery courtesy of Planet Lab PBC).

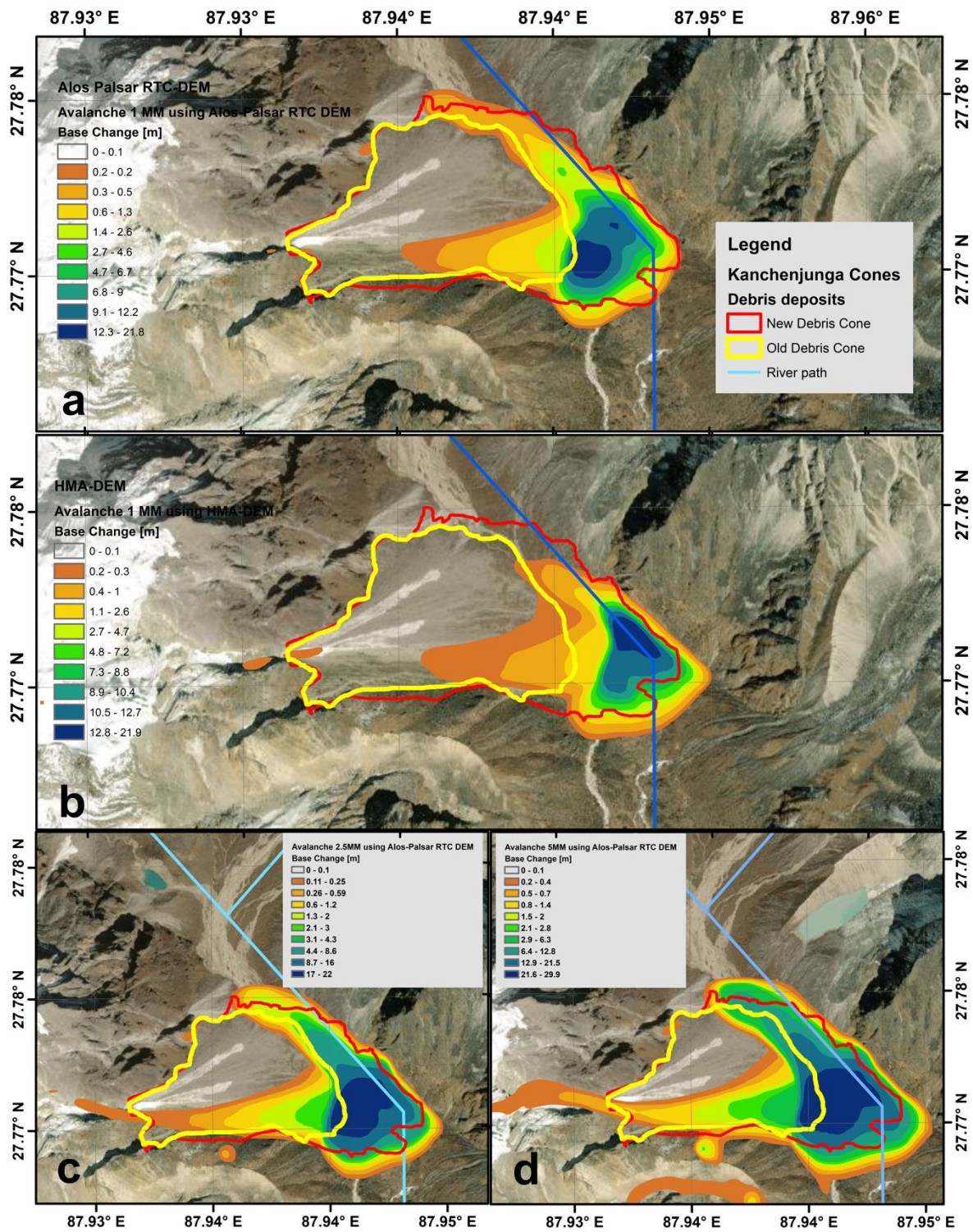

**Figure 3. Base change modeled with R.Avaflow for three different avalanche volumes: (a) 1 x10<sup>6</sup> m³ using an**
**ALOS PALSAR RTC DEM, (b) 1 x10⁶ m³ using the void filled HMA-DEM, (c) 2.5 x10⁶ m³ (bottom left) using an**
**ALOS PALSAR RTC DEM, and (d) 5 x10⁶ m³ (bottom right) using an ALOS PALSAR RTC DEM. Of the three**
**estimates and 2 DEMs, 1 x 10⁶ m³ using an ALOS PALSAR RTC DEM most consistently matched the extent and**
**depth of the new debris cone deposited in August 2022 (red line; the yellow line represents the extent of the**
**historic debris cone).**
floodwater diversion mechanisms, such as the rock-filled gabion walls currently protecting tourist
lodges in the Mt. Everest region (e.g., Rounce et al. 2017; Byers et al. 2022) using participatory
processes as outlined in Watanabe et al. (2016).
**5  Conclusion**
Beginning in 2020, a series of small-to-medium, torrent-like pulses commenced upon a historic debris
cone located approximately 2 km down valley from the lake, culminating in a relatively large
avalanche event that occurred sometime between 16 and 21 August 2022. The August 2022 event
deposited debris with an area of 0.6 km$^2$ and estimated volume in the order of $10^6$ m$^3$. No fatalities
from the event occurred because of the absence of humans and livestock in the vicinity when the
event occurred. Likewise, no impoundment of the Nupchu Khola, and formation of a potentially
dangerous backwater lake, occurred as a result of debris blockage, although such scenarios happen
routinely in high mountain environments.
The improvement of remote area event reporting mechanisms, especially to authorities in the capital,
Kathmandu, could help with the development of hazard mitigation technologies and response.
Likewise, more systematic monitoring of cryospheric events by scientists, using remote sensing
platforms and hazard mapping tools, could help with the development of more effective early warning
systems for vulnerable communities, livestock, and adventure tourists. Ultimately, this could lead to a
minimization of losses and damage due to multi-hazard events.

*Data availability.* Declassified KH-9 Hexagon satellite imagery is available at
https://earthexplorer.usgs.gov/. Planet Dove and SuperDove satellite imagery is available
at https://www.planet.com/explorer/. The ALOS PALSAR Radiometric Terrain Corrected high-res DEM
"AP_13152_FBD_F0540_RT1" is available at https://search.asf.alaska.edu/. The High Mountain Asia 8-
meter DEM is available at https://nsidc.org/data/hma_dem8m_ct/versions/1#anchor-1
*Author contributions*. ACB conceived the study and wrote the original narrative, with contributions
from MS-V, DS, DM, MBC, and RA. MBC created Figure 1, DS and RA created Figure 2, and MS-V
created Figure 3. MS-V conducted the numerical simulations of avalanche volumes shown in Figure 3.
All authors revised and contributed to the final manuscript.
*Competing Interests*. The contact author has declared that none of the authors has any competing
interests.
*Acknowledgements*. The Fulbright Nepal Scholar Program is thanked for its support of A.C. Byers
during his six-month field study of contemporary impacts on alpine ecosystems in the Kanchenjunga
Conservation Area, eastern Nepal. The Department of National Parks and Wildlife Conservation,
Kanchenjunga Conservation Area, Department of Geography at Tribhuvan University are also thanked
for their interest in and support of the project. Support for M. Somos-Valenzuela during the
preparation of this paper was provided by the Chilean Science Council (ANID) through the Program of
International Cooperation (PII-180008). Support for D.H. Shugar was provided by the Natural Sciences
and Engineering Research Council of Canada (DG-2020-04207) and Alberta Innovates. Support for D.
McGrath was provided by NASA award 80NSSC20K1343.

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
