# Peer review of "Brief Communication: An Ice-Debris Avalanche in the Nupchu Valley, Kanchenjunga Conservation Area, Eastern Nepal"

_EGUsphere, 2023_

## Community Comment (CC1)

Fig2: is there no suitable satellite image closer after the event? Only several months later?

*Indeed, the number of cloud-free images from after the event was disappointing. There are images from Aug 21, 24, 25, Sept 21, Oct 3, 8, with clouds obscuring the source area, as well as some where fresh snowfall (e.g. Oct 16) makes interpretation trickier. While we could map the distribution of the debris sheet from the earlier imagery (e.g. Aug 21), we felt it better to show a later image in which the source zone (and the changes therein) is easier to see. I have attached an example image from Aug 24.*

[Figure]

*Example Planetscope image from Aug 24*

---

## Author Response (AR1)

*REVIEWER #1*
*COMMENTS AND AUTHOR RESPONSES*
*28 SEPTEMBER 2023*

*Thank you for your very helpful comments on our Brief Communication. Our responses to your various points are shown in italics below:*

The brief communication describes and interprets ice-avalanche deposits from a site in Nepal. The contribution also concludes in recommendations related to monitoring and hazard risks.

My main concern with this study are the risk assessments and monitoring recommendations that are, first, not really careful, or detailed enough  grounded, and second, not up to scientists, and especially not in a brief communication without space for sound and detailed assessments. I acknowledge the good will of the authors, but given the pressure or confusion such publication could put on local authorities or damage it could do (e.g. impact on tourism and thus local income), such paper should not be published in TC. In an extreme case it could result in liability issues for the involved parties. A number of details suggest this submission was not done carefully, see below details.

- *We agree that this is not a particularly unique event (i.e., it is one of several that may occur annually in the region), but such events are not commonly reported and are rarely if ever documented.*  *We have also revised the paper to focus entirely on the event itself, with brief mention of possible mitigation measures confined to the Discussion section. Please consult the track changes version of our edits for further details.*

~~*Likewise, and based on the collective authors' decades of experience working in the field of cryospheric hazards, it is precisely the lack of such information that is part of the problem facing governments and practitioners face in the design and implementation of effective mitigation measures. As mentioned in the Introduction, although modern technologies have enabled a more rapid identification of similar events, "…. remain unreported because of their remoteness, inaccessibility, poor communications, and/or absence of people," the current event an excellent case in point. If published, the paper would be shared immediately with KCA authorities and local communities, including the recommendations that villages such as Kampuchen begin thinking seriously about installing flood mitigation technologies such as the gabions mentioned in the paper. No pressure on local authorities, or negative impacts on the region's tourism, are envisioned, rather the exact opposite in the form of providing information and recommendations in a data deficient region is fully expected (and has been the case in numerous other community-based projects*~~

I don't understand what the purpose of this paper is? It looks more like a blog (with attention to my above concerns, though). What is special with this event? The paper contains and concludes several little explained and discussed but strong statements (undagerous lake, recommendation of early warning and monitoring ...). If I don't overlook a non-searchable part of the submission, there is a substantial number of references in the reference list that I don't find in the text, and vice-versa. This adds to my impression that this submission was not done carefully enough.

***Thank you again for pointing out several areas where the paper could be strengthened. We*** *are currently preparing a* ***have now prepared a*** ***revised version of the paper which we feel will address*** *most* ***all*** ***of your concerns*** ***by its focus primarily on the event itself, with other aspects such as possible mitigation measures for local villages confined to the Discussion section only. References have also been checked and are now complete.*** *, including greater clarity describing the importance of the event, the importance of informing local people of the current status of Nupchu glacial lake, the importance of the recommendations, etc.*

 Detail comments:

line 24: this straight and extensive monitoring advice (half of the abstract) is not up to scientists without mandate by the responsible authorities.

***References to monitoring have been removed from the abstract.***

*We disagree in part. One of the many challenges currently facing the development of effective responses to the impacts of climate change is the lack of communication between scientists and government (we assume this is what you mean by "responsible authorities"). Of course, for science to have the greatest impact, it must be accepted by local authorities and the public. But if scientists were to always wait for the mandates of "responsible authorities" many such events would go unreported. Likewise, scientists who communicate only with scientists accomplish little of use by the communities and stakeholders impacted most by such events as those described in the paper. Please see Watanabe et al. 2016 in the paper's references for further detail.*

62: I don't understand what the biological richness of the area has to do with the topic of the paper, ice avalanche risks.

***Removed from the narrative.***

*We agree that this sentence, as written, is not justified appropriately, and have thus deleted it. However, high biodiversity is one of the most promising assets that the KCA has in terms of its development of adventure tourism, which in turn can be impacted by climate change and catastrophic events.*

90: Why specifically this DEM, not e.g. the HMA DEM?

***Thank you for this suggestion. As several of the co-authors agree that HMA DEM would have been the better choice, we accessed the data and mapped de DEM realizing that in the study area there are several patches with no data as you can see in the Figure below. Theses discontinuities in the data make the HMA DEM not suitable for the debris flow simulation. Therefore, we decided to continue using the DEM originally selected (ALOS-PALSAR) since according to the literature, it has the best resolution and elevation accuracy for mountain and rugged tarrains from the freely available DEMS in the study area.***

Bhardwaj, A. (2019). Assessment of Vertical Accuracy for TanDEM-X 90 m DEMs in Plain, Moderate, and Rugged Terrain. The 2nd International Electronic Conference on Geosciences, 8. https://doi.org/10.3390/IECG2019-06208

Shean, D. (2017). High Mountain Asia 8-meter DEM Mosaics Derived from Optical Imagery, Version 1 [Data Set]. Boulder, Colorado USA. NASA National Snow and Ice Data Center Distributed Active Archive Center. https://doi.org/10.5067/KXOVQ9L172S2. Date Accessed 09-25-2023.

Shawky, M., Moussa, A., Hassan, Q. K., & El-Sheimy, N. (2019). Pixel-based geometric assessment of channel networks/orders derived from global spaceborne digital elevation models. Remote Sensing, 11(3). https://doi.org/10.3390/rs11030235

[Figure]

***Figure XXXX: Discontinuity of the HMA DEM in the area of the debris flow*** 119: frequency-magnitude information is just given in the figure caption. Information, more detailed, is needed in the main text.

*We propose adding the following into the main text, starting at line 119: "Time series satellite images showing the periodic occurrence of surficial debris flows upon the original deposition. These appear to have accelerated in both frequency in magnitude beginning several years ago, leading up to the main event that occurred between 16 and 21 August 2022."*

Fig2: is there no suitable satellite image closer after the event? Only several months later?

*Indeed, the number of cloud-free images from after the event was disappointing. There are images from Aug 21, 24, 25, Sept 21, Oct 3, 8, with clouds obscuring the source area, as well as some where fresh snowfall (e.g. Oct 16) makes interpretation trickier. While we could map the distribution of the debris sheet from the earlier imagery (e.g. Aug 21), we felt it better to show a later image in which the source zone (and the changes therein) is easier to see. I have attached an example image from Aug 24.*

[Figure]

*Example Planetscope image from Aug 24*

around 153: This is quite far-fetched based on a few satellite images and one avalanche event.

*We propose to revise this sentence as "…These can be expected to increase in frequency as well as magnitude…." to "If these increase in frequency as well as magnitude in the coming decades within the Kanchenjunga region, they could include such events as new GLOFs, englacial conduit floods, rockfall-induced rock avalanches, and other phenomena."*

around 159: Is there more risk than in the many other Himalayan valleys? Such avalanches might have happened at several other places. Are you sure this is a special event? What are your arguments for that?

*It is one of many such events that occur annually in the Himalayas which are neither reported to central government authorities nor studied by scientists in both the field and laboratory. For example, only one GLOF event was in ICIMOD's records of GLOFs in the Kanchenjunga region of Nepal in 2019, when a subsequent field, lab, and oral testimony investigation revealed that 8 major GLOF events had occurred. We feel that our Brief Communication describing this event in the Kanchenjunga region will be of use to the GON's Department of Hydrology and Meteorology, ICIMOD, USAID, and other climate change within the Hindu Kushi-Himalayan region, including threatened villages such as Kampuchen.*

164: You cannot mention in the conclusions, without any assessment presented in the paper, that a certain lake represents no outburst risk.

**

*Mention of Nupchu as a potential danger has been omitted from the conclusion, which now focuses on the actual event itself.*

around 184: These are pretty wide conclusions based on one local event.

*~~Thank you for this comment. We agree that, as written, these conclusions are quite broad. However, they were also facilitated and augmented by decades of community-based field and laboratory projects in cryospheric hazards and their mitigation measures in the Himalayas and Andes, including previous work by various authors of the paper. Given the space restrictions, a more lengthy discussion is not possible, but we have added an additional clause pointing to the existing body of literature. but we feel that the present revised paper addresses this concern as well.~~*

*The conclusions now focus entirely on the event itself, thanks to the thoughtful comments of Reviewer #1. They now read:*

Beginning in 2020, a series of small-to-medium, torrent-like pulses commenced upon a historic debris cone located approximately 2 km down valley from the lake, culminating in a relatively large avalanche event that occurred sometime between 16 and 21 August 2022. The August 2022 event deposited debris with an area of 0.6 km$^2$ and estimated volume in the order of $10^6$ m$^3$. No fatalities from the event occurred because of the absence of humans and livestock in the vicinity when the event occurred. Likewise, no impoundment of the Nupchu Khola, and formation of a potentially dangerous backwater lake, occurred as a result of debris blockage, although such scenarios happen routinely in high mountain environments.

The improvement of remote area event reporting mechanisms, especially to authorities in the capital, Kathmandu, could help with the development of hazard mitigation technologies and response. Likewise, more systematic monitoring of cryospheric events by scientists, using remote sensing platforms and hazard mapping tools, could help with the development of more effective early warning systems for vulnerable communities, livestock, and adventure tourists. Ultimately, this could lead to a minimization of losses and damage due to multi-hazard events.

Brief Communication: An Ice-Debris Avalanche in the Nupchu Valley, Kanchenjunga Conservation

Area, Eastern Nepal

*REVIEWER #2*

*COMMENTS AND AUTHOR RESPONSES*

*28 SEPTEMBER 2023*

*Authors Response: Thank you very much for your thoughtful and helpful comments. We have provided our responses below each of the key points below:*

Comment: This short paper describes a mass movement event in the Kangchenjunga Conservation Area.  It is fairly interesting but appears to be a bit underwhelming for a mainstream journal like *The Cryosphere*.  Even the Abstract describes the events as "a series of small-to-medium, torrent-like pulses" and "a comparatively large ice-debris avalanche event".  Neither description suggests that the events were particularly remarkable and this is supported by the fact that the events failed to impact local infrastructure and inhabitants.  As a result, although the paper is a quite interesting if short study, there is quite a lot that could have been added to this such as photographs from the site itself and, perhaps some meteorological observations or climate data.  The paper could also have included some detailed risk assessments for the local village.

*Response: We agree that the topic is worthy of a more comprehensive study in the future, which could include more narrative, photographs, meteorological and climate data.  However, in the case of our Brief Communication, our primary objective is to highlight this style of event in this region, which has not received extensive study previously. We are constrained by the "Brief Communications" requirements: "Brief communications are timely, peer-reviewed, and short (2–4 journal pages). These may be used to (a) report new developments, significant advances, and novel aspects of experimental and theoretical methods and techniques which are relevant for scientific investigations within the journal scope; (b) report/discuss on significant matters of policy and perspective related to the science of the journal, including "personal" commentary; and (c) disseminate information and data on topical events of significant scientific and/or social interest within the scope of the journal. Brief communications have a maximum of 3 figures and/or tables, a maximum of 20 references, and an abstract length not exceeding 100 words. The manuscript title must start with "Brief communication:". Again, we thank you for your comments and will consider incorporating them in a future longer-format study.*

Comment: Is there any data on the magnitude/frequency relationships of such events in this region of the Himalayas?

*Response: There is a lack of information related to the magnitude/frequency of such events in the Kanchenjunga region. In fact, this study is one of only a handful of peer reviewed papers available for the region that are concerned with high magnitude/low frequency events. We feel that this situation only underscores the importance of making the information in the current paper available to a wider audience, even as a short-format Brief Communications.*

Comment: What is the link between this and regional climate change? This is discussed briefly but this could be significantly expanded.

and…

Comment: The paper could make quite a nice local site example in a regional journal, but as it stands, I do not think that the paper is significant enough for TC.

*Response: Thank you for these suggestions. These are important points that are worthy of further investigation but given the limitations imposed by the Brief Communications format, we are unable to expand the analysis further.*

*Thanks you again for taking the time to provide feedback on our Brief Communications submission to The Cryosphere.*

Specific comments.

Line 67 Presumably glaciers have been receding since before then….Last Glaciation?

- *The sentence has been modified and expanded as follows:*

Valley glaciers are largely debris-covered and have been receding since the most recent maximum during the Little Ice Age. Hooker (1854), for example, wrote in 1849 of observing glacial moraines that provided proof "…of glaciers having once descended to from 8,000 to 10,000 feet in every Sikkim and east Nepal valley…" (Hooker 1854: 166). The British alpinist Freshfield (1903: 236) writes of the "glacial shrinkage" he encountered in the Lhonak region in 1899, as well as throughout both the Nepal and Sikkim sides of the Kanchenjunga massif. Although the Kanchenjunga region received some of the earliest study, exploration, and mountaineering expeditions in Nepal by outsiders (Thapa 2009), relatively little glacier and cryospheric hazards research has been conducted to date. For example, until 2019 only one GLOF event was on record for the region (Watanabe et al. 1998; ICIMOD 2011), although subsequent research revealed that at least seven others had occurred since 1921 (Byers et al. 2020). The Nupchu valley, where the ice-debris avalanche of concern occurred, is

used seasonally for yak herding, potato farming, and tourism, with four operational tourist lodges in the village of Kampuchen as of the fall of 2022 (Figure 1).

Lines 149-154  Rather vague assessment.  The relationships between such events and climate warming not always apparent, and this requires a more critical assessment.  The link between climate change and GLOFs is not clear (see papers by Georg Veh).

- ***The sentence has been revised as follows:*** Still, the acceleration of torrent-like pulses of debris upon the historic debris cone since 2020 suggests that they could have been linked to contemporary warming trends, similar to larger-scale mass wasting events found elsewhere in the Himalaya (e.g., Shugar et al. 2021; Kääb et al. 2021; Taylor et al. 2023). If these increase in frequency as well as magnitude in the coming decades within the Kanchenjunga region, they could include such events as new GLOFs, englacial conduit floods, rockfall-induced rock avalanches, and other phenomena (e.g., Byers et al. 2017, 2022).Vulnerable villages, such as Kampuchen, may wish to consider the installation of preventative floodwater diversion mechanisms, such as the rock-filled gabion walls currently protecting tourist lodges in the Mt. Everest region (e.g., Rounce et al. 2017; Byers et al. 2022) using participatory processes as outlined in Watanabe et al. (2016).

Several papers in the text are not listed in the reference list, and vice versa.

***The references within the text and in the reference list have been corrected. Please consult the track changes versi***

---

## Author Response (AR2)

**University of Colorado at Boulder**

**Institute of Arctic and Alpine Research**

Campus Box 450
Boulder, Colorado 80309-0450

Alton C. Byers, Ph.D.
Senior Research Scientist
Office phone: (304) 636-6980
Cell phone: (571) 481-8650
alton.byers@colorado.edu

27 October, 2023

Dear Editor and Reviewers,

Thank you for your recent communications and comments regarding our paper, *Brief Communication: An Ice-Debris Avalanche in the Nupchu Valley, Kanchenjunga Conservation Area, Eastern Nepal.*
Your concerns and our responses are shown below:

**While Reviewer #1 is now satisfied with the revised manuscript, I share the concerns of Reviewer #2 that:**
**1. The paper still lacks consideration of how typical such events could be in this region.**

To address Reviewer #2's request to include information regarding the (a) frequency/magnitude of events such as the Nupchu ice/debris flow, and (b) their relation to climate change, we have revised the following paragraph under the Discussion section, starting at line 144, as follows:

> *Still, the acceleration of torrent-like pulses of debris upon the historic debris cone since 2020 suggests that these events may be linked to contemporary warming trends, similar to those that may have triggered larger-scale mass wasting events elsewhere in the Himalaya (e.g., Shugar et al. 2021; Kääb et al. 2021; Taylor et al. 2023). The frequency of such ice-debris flow events within the KCA region, and more broadly across the Himalaya, is unknown. However, with projections of continued warming in these regions (e.g., Lalande et al. 2021), a more systematic approach to determining their historic frequency, as well as a better understanding of their triggers, is warranted. After further evaluation, vulnerable villages, such as Kampuchen, may wish to consider the installation of preventative floodwater diversion mechanisms, such as the rock-filled gabion walls currently protecting tourist lodges in the Mt. Everest region (e.g., Rounce et al. 2017; Byers et al. 2022) using participatory processes as outlined in Watanabe et al. (2016).*

Note that the paragraph above also includes a new reference (i.e., Lalande et al. 2021) to climate change trends in the High Mountain Asia region as well.

**2. The tracked changes and lack of clarity in some aspects of the response letter (for example, Figure xxxx) give the impression that careful review of the materials was not completed before submission.**

We apologize if the response to and figure for the paper's selection of DEM model was not entirely clear. We have revised the response to now read as follows:

90: Why specifically this DEM, not e.g. the HMA DEM?

***Thank you for this suggestion. Our initial simulation was conducted on the ALOS-PALSAR DEM since prior research (e.g., Bhardwaj, 2019; Shawky et al. 2019) suggested it had the best resolution and elevation accuracy in mountainous and rugged terrain. Given this suggestion, we investigated the use of the HMA DEM (Shean 2017) but found that there are several data gaps (Figure 1) within our simulation area that would require data interpolation to make it suitable for study. As we do not expect significant sensitivity in the model results to modest changes in the input DEM, we feel that the use of the original DEM is sufficient.***

Bhardwaj, A. (2019). Assessment of Vertical Accuracy for TanDEM-X 90 m DEMs in Plain, Moderate, and Rugged Terrain. The 2nd International Electronic Conference on Geosciences, 8. https://doi.org/10.3390/IECG2019-06208

Shean, D. (2017). High Mountain Asia 8-meter DEM Mosaics Derived from Optical Imagery, Version 1 [Data Set]. Boulder,

Shawky, M., Moussa, A., Hassan, Q. K., & El-Sheimy, N. (2019). Pixel-based geometric assessment of channel networks/orders derived from global spaceborne digital elevation models. Remote Sensing, 11(3). https://doi.org/10.3390/rs11030235

[Figure]

*Figure 1: HMA DEM elevations [m] in the vicinity of the debris flow. Data gaps in the model domain would require interpolation in order to be suitable for simulations.*

Thank you again for kind attention to and interest in our Brief Communications submission. Please feel free to contact me at any time if there are any remaining concerns or questions.

Sincerely,

Alton

Alton C. Byers, Ph.D.
2023-2026 Fulbright Specialist (Global)
Fulbright Nepal Research Scholar, Kanchenjunga Conservation Area, eastern Nepal (June-November 2022)
Distinguished Visiting Scholar, College of the Environment, Wesleyan University, Middletown, CT (2021-2022)

Faculty Research Scientist, Senior Research Associate, Institute for Arctic and Alpine Research (INSTAAR), University of Colorado at Boulder

Explorer, National Geographic Society, and Expert, National Geographic Expeditions

Senior Fellow, The Mountain Institute, Lima, Peru

Address: 406 Westridge Drive, Elkins, WV 26241

Tel: 304-636-6980 (landline office)

US Cell: 571-481-8650

Nepal Cell: 974-566-9921

Websites:

https://instaar.colorado.edu/people/alton-c-byers/

https://instaar.colorado.edu/research/programs/himap/

https://www.nationalgeographic.org/find-explorers/alton-c-byers-iii

http://mountain.org/about-us/experts-advisors/

---

## Author Response (AR3)

12 December, 2023

Dear Editor and Reviewers,

Thank you for your recent communications and comments regarding our paper, *Brief Communication: An Ice-Debris Avalanche in the Nupchu Valley, Kanchenjunga Conservation Area, Eastern Nepal*.

Your comments, and our responses, are shown below:

- **1. (Reviewer) The paper improved as it now focusses on the event which is the main topic of the contribution. However, the paper still lacks reference on how typical such events could be. The abstract states widespread distribution of such events, but no further discussion or assessment is given later in the paper. Rather the paper refers to other types of high mountain hazards that the paper doesn't study. The authors would need to improve the general significance of the study based on discussing or assessing the same type of events as described in the paper.**

- **(Authors)** To address Reviewer #2's request to include information regarding the (a) frequency/magnitude of events such as the Nupchu ice/debris flow, and (b) their relation to climate change, we have revised the following paragraph under Discussion, starting at line 144, as follows:

  *Still, the acceleration of torrent-like pulses of debris upon the historic debris cone since 2020 suggests that these events may be linked to contemporary warming trends, similar to those that may have triggered larger-scale mass wasting events elsewhere in the Himalaya (e.g., Shugar et al. 2021; Kääb et al. 2021; Taylor et al. 2023). The frequency of such ice-debris flow events within the KCA region, and more broadly across the Himalaya, is unknown. However, with projections of continued warming in these regions (e.g., Lalande et al. 2021), a more systematic approach to determining their historic frequency, as well as a better understanding of their triggers, is warranted. After further evaluation, vulnerable villages, such as Kampuchen, may wish to consider the installation of preventative floodwater diversion mechanisms, such as the rock-filled gabion walls currently protecting tourist lodges in the Mt. Everest region (e.g., Rounce et al. 2017; Byers et al. 2022) using participatory processes as outlined in Watanabe et al. (2016).*

[Note that the paragraph above also includes a new reference (i.e., Lalande et al. 2021) to climate change trends in the High Mountain Asia region as well].

- **2. (Reviewer) I see that the HMA DEM has some voids over the study area, but these seem not massive and could be interpolated (from what is visible in the figure). The ALOS DEM has also been processed and might have been void-filled that way. A brief comparison of results using the ALOS and HMA DEMs could be useful to investigate the sensitivity of the modelling to the DEM used. I would be happy to learn that the difference is minor, but many readers could wonder the same (as the co-author discussion confirms).**

- **(Authors)** We ran a R.Avaflow model using the HMA-DEM that was void filled in ArcGIS, and the same parameters used in the original model for an avalanche of 1 MM. We then visually

compared the debris cone deposit observed from optic images with this new result, and they look, generally, in good agreement.

However, as is shown in Figure 1 below, we can see that the results from the Alos Palsar RTC DEM are in better agreement with the delineation of the debris cone. It seems that the flow modeled using the HMA-DEM develops faster which ends up extending the debris cone deposit in the main direction of the flow having some uphill runout in an area where the optic images do not show debris cone.

[Figure]

- Figure 1: Comparison of the deposit using the ALOS PALSAR Radiometric Terrain Corrected high-res (Above) and the HMA-DEM (below). The delineation of the actual cone is superimposed to visualize the differences.

- **3. (Reviewer) I am still puzzled about the "ALOS PALSAR DEM" used. According to the cited ASF there is no such DEM. The ASF PALSAR data contain other DEM data for reference. Are you referring to this?**
  **https://asf.alaska.edu/data-sets/derived-data-sets/alos-palsar-rtc/alos-palsar-radiometric-terrain-correction/**
  **Or the ALOS PRISM DEM by JAXA? Or else?**

(Authors) We appreciate the reviewer's comment which allows us to clarify in more detail the DEM used in the text. As is mentioned by the reviewer and in the first reference in the list of references, we are referring to an ALOS PALSAR Radiometric Terrain Corrected high-res product as shown in the figure below. When we did the search in the https://search.asf.alaska.edu/ website we only indicated ALOS PALSAR as the dataset of interest (see figure below). As a result, in the paper we thought it only necessary to include that information to allow the readers to find the DEM used. However, we accept the reviewer's comment which clarifies further the datasets

used. In the paper at line 93 to 94, we now indicate that we are referring to an ALOS PALSAR Radiometric Terrain Corrected high-res file. We have also included this information in the Data Availability section as *"The ALOS PALSAR Radiometric Terrain Corrected high-res DEM "AP_13152_FBD_F0540_RT1" used for the R.Avaflow is available at https://search.asf.alaska.edu/."*

- The instruction for the citation can be found at: https://asf.alaska.edu/data-sets/sar-data-sets/alos-palsar/alos-palsar-how-to-cite/

[Figure]

Figure 2: Screen shot of the https://search.asf.alaska.edu/ where the DEM was downloaded on February 11, 2023.

- **4. (Reviewer) You write that you compare modelled deposition depths to field data but don't describe field investigations of deposit depths. Line 144 in the ATC.**

- **(Authors)** On line 82/93, in the Methods section, the following clarification has been added: regarding the use of a Nikon Forestry Pro Rangefinder to determine depths of the ice-debris avalanche:

  *Field-based observations and assessments of Nupchu Pokhari (glacial lake), other nearby lakes, and the ice-debris avalanche were conducted between 1–20 September 2022. Methods included GPS-based route mapping, photography of avalanche features, oral testimony, and literature reviews. A Nikon Forestry Pro Rangefinder was used to determine the depth of ice/debris deposits where the Nupchu river had incised deposits down to the original streambed. Historic ........*

- **5. (Reviewer) Fig XXXX in the Author response is not explained and rather contains an open to-do item**

- **(Authors)** Our initial simulation was conducted on the ALOS-PALSAR DEM since prior research (e.g., Bhardwaj, 2019; Shawky et al. 2019) suggested it had the best resolution and elevation accuracy in mountainous and rugged terrain. Given this suggestion, we investigated the use of the HMA DEM (Shean 2017) but found that there are several data gaps (Figure 1) within our simulation area that would require data interpolation to make it suitable for study. As we do not expect significant sensitivity in the model results to modest changes in the input DEM, we feel that the use of the original DEM is sufficient.

Bhardwaj, A. (2019). Assessment of Vertical Accuracy for TanDEM-X 90 m DEMs in Plain, Moderate, and Rugged Terrain. The 2nd International Electronic Conference on Geosciences, 8. https://doi.org/10.3390/IECG2019-06208

Shean, D. (2017). High Mountain Asia 8-meter DEM Mosaics Derived from Optical Imagery, Version 1 [Data Set]. Boulder, Colorado USA. NASA National Snow and Ice Data Center Distributed Active Archive Center. https://doi.org/10.5067/KXOVQ9L172S2. Date Accessed 09-25-2023.

Shawky, M., Moussa, A., Hassan, Q. K., & El-Sheimy, N. (2019). Pixel-based geometric assessment of channel networks/orders derived from global spaceborne digital elevation models. Remote Sensing, 11(3). https://doi.org/10.3390/rs11030235

[Figure]

***Figure 3: HMA DEM elevations [m] in the vicinity of the debris flow. Data gaps in the model domain would require interpolation in order to be suitable for simulations.***

- **6. (Reviewer) The discussion comments between co-authors in the ATC are unusual to submit but could be viewed helpful. Though, these comments and some open do-todo's in the response letter (and ATC?) give the impression that the items have been re-submitted before carefully finalizing them.**

- **(Authors)** We apologize if the discussion comments created the impression of lacking careful consideration before finalization. In fact, all authors found the candid conversations to be most helpful in refining the manuscript before its re-submission.

- **7. (Reviewer) The fact that I can still not find all text references in the reference list confirms this impression (ICIMOD, WWF, Rounce ...).**

- **(Authors)** All missing references have now been added to the Reference list.

Thank you again for kind attention to and interest in our Brief Communications submission. Please feel free to contact me at any time if there are any remaining concerns or questions.

Sincerely,

Alton C. Byers, Ph.D.

---

## Author Response (AR4)

20 December, 2023

**Detailed Response to Reviewers and Editor:**

**(Reviewer):** *In my view, the paper is now acceptable for publication, pending one strong recommendation:*
*Using a compiled DEM for avalanche modelling (the DEM layer of ASF PALSAR data), the roots of which the authors are not able to find, should be avoided in the sense of open and reproducible science. As the DEM used is not an original DEM, ASF may change it at any time. At the same time, I thank the authors to present a sensitivity test based on the (original) HMA DEM. Taken both together, I strongly recommend to include Fig 1 of their response (the ALOS vs. HMA-DEM comparison) in the final paper together with a few words of explanation, or in a Supplement. I leave to the editor whether the Figure can be added to the main paper (preferred), or needs to go to a Supplement for space reasons.*

Thank you very much for this recommendation. Additional panels showing a comparison of the deposit from the two different DEMs have been added to Figure 3 (please see revised manuscript). Figure 3's caption now reads:

**Figure 3. Base change modeled with R.Avaflow for three different avalanche volumes: (a) 1 x10$^6$ m$^3$ using an ALOS PALSAR  RTC DEM, (b) 1 x10$^6$ m$^3$ using the void filled HMA-DEM, (c) 2.5 x10$^6$ m$^3$ (bottom left)  using an ALOS PALSAR  RTC DEM, and (d) 5 x10$^6$ m$^3$ (bottom right) using an ALOS PALSAR  RTC DEM. Of the three estimates and 2 DEMs, 1 x 10$^6$ m$^3$ using an ALOS PALSAR  RTC DEM most consistently matched the extent and depth of the new debris cone deposited in August 2022 (red line; the yellow line represents the extent of the historic debris cone).**

Associated edits include line 92, which now reads as:

For the terrain elevation, two DEMs were used: the ALOS PALSAR Radiometric Terrain Corrected high resolution 12.5 m DEM  (AP_13152_FBD_F0540_RT1) (ASF DAAC 2014) and the High Mountain Asia 8m resolution DEM (Shean, 2017), that was void filled using the Elevation Void Fill function in ArcGIS 10.8.

And line 123:

Of the three different volume estimates tested (1, 2.5, and 5 million m$^3$) using two DEMs and R.Avaflow, an avalanche volume of 1 x 10$^6$ m$^3$ using an ALOS PALSAR  RTC DEM most consistently matched the extent (red line) and depth of the new debris cone deposited as determined by our field observations (Figure 3).

On line 19, Data Availability now reads as:

*Data availability.* Declassified KH-9 Hexagon satellite imagery is available at https://earthexplorer.usgs.gov/. Planet Dove and SuperDove satellite imagery is available at https://www.planet.com/explorer/. The ALOS PALSAR Radiometric Terrain Corrected high-res DEM "AP_13152_FBD_F0540_RT1" is available at https://search.asf.alaska.edu/. The High Mountain Asia 8-meter DEM is available at https://nsidc.org/data/hma_dem8m_ct/versions/1#anchor-1

In the References, a new related reference is cited as:

Shean, D. (2017). High Mountain Asia 8-meter DEMs Derived from Cross-track Optical Imagery. Boulder, CO: NASA NSIDC DAAC: NASA National Snow and Ice Data Center Distributed Active Archive Center. https://doi.org/10.5067/0MCWJJH5ABYO. Accessed on July 20, 2023.

Thank you again for your valuable comments and guidance throughout the preparation of our paper. We hope that the above changes will now certify the paper as suitable for publication in *The Cryosphere*.

All the best,

Alton C. Byers, Ph.D.
2023-2026 Fulbright Specialist (Global)
Faculty Research Scientist, Senior Research Associate, Institute for Arctic and Alpine Research (INSTAAR), University of Colorado at Boulder
Explorer, National Geographic Society, and Expert, National Geographic Expeditions
Senior Fellow, The Mountain Institute, Lima, Peru
Address: 406 Westridge Drive, Elkins, WV 26241
Tel: 304-636-6980 (landline office)
US Cell: 571-481-8650
Nepal Cell: 974-566-9921
Websites:
https://instaar.colorado.edu/people/alton-c-byers/
https://instaar.colorado.edu/research/programs/himap/
https://www.nationalgeographic.org/find-explorers/alton-c-byers-iii
http://mountain.org/about-us/experts-advisors/

---

## Author Response (AR5)

27 December 2023

Dear Chris,

Thanks very much for all of your help toward the final acceptance of our manuscript for publication in the *Cryosphere*. As very little has been published about glacial and climate change processes in the Kanchenjunga region of Nepal, we hope that the paper will facilitate an increased understanding of contemporary and future cryospheric hazards in the region, as well as government- and village-based mitigation opportunities.

All the best,

Alton C. Byers, Ph.D.
2023-2026 Fulbright Specialist (Global)
Faculty Research Scientist, Senior Research Associate, Institute for Arctic and Alpine Research (INSTAAR), University of Colorado at Boulder
Explorer, National Geographic Society, and Expert, National Geographic Expeditions
Senior Fellow, The Mountain Institute, Lima, Peru
Address: 406 Westridge Drive, Elkins, WV 26241
Tel: 304-636-6980 (landline office)
US Cell: 571-481-8650
Nepal Cell: 974-566-9921
Websites:
https://instaar.colorado.edu/people/alton-c-byers/
https://instaar.colorado.edu/research/programs/himap/
https://www.nationalgeographic.org/find-explorers/alton-c-byers-iii
http://mountain.org/about-us/experts-advisors/